# Sensing complementary temporal features of odor signals enhances navigation of diverse turbulent plumes

**Viraaj Jayaram**[1,2,3†], **Nirag Kadakia**[2,3†], **Thierry Emonet**[1,2,3*]

[1]Department of Physics, Yale University, New Haven, United States; [2]Department of Molecular, Cellular and Developmental Biology, Yale University, New Haven, United States; [3]Quantitative Biology Institute, Yale University, New Haven, United States

**Abstract:** We and others have shown that during odor plume navigation, walking *Drosophila melanogaster* bias their motion upwind in response to both the frequency of their encounters with the odor (Demir et al., 2020) and the intermittency of the odor signal, which we define to be the fraction of time the signal is above a detection threshold (Alvarez-Salvado et al., 2018). Here, we combine and simplify previous mathematical models that recapitulated these data to investigate the benefits of sensing both of these temporal features and how these benefits depend on the spatio-temporal statistics of the odor plume. Through agent-based simulations, we find that navigators that only use frequency or intermittency perform well in some environments – achieving maximal performance when gains are near those inferred from experiment – but fail in others. Robust performance across diverse environments requires both temporal modalities. However, we also find a steep trade-off when using both sensors simultaneously, suggesting a strong benefit to modulating how much each sensor is weighted, rather than using both in a fixed combination across plumes. Finally, we show that the circuitry of the *Drosophila* olfactory periphery naturally enables simultaneous intermittency and frequency sensing, enhancing robust navigation through a diversity of odor environments. Together, our results suggest that the first stage of olfactory processing selects and encodes temporal features of odor signals critical to real-world navigation tasks.

**\*For correspondence:**
thierry.emonet@yale.edu

†These authors contributed equally to this work

## Editor's evaluation

This article by Jayaram and colleagues uses computational modeling approaches to examine how temporal filtering of an odor signal contributes to navigation success in different odor environments. The article advances the literature in considering how different algorithms may be optimal for different environments. The provided evidence suggests an intriguing trade-off between frequency and 'intermittency' sensing.

## Introduction

The complexity of natural odor plumes makes olfactory navigation a difficult task. Turbulent flows produce rapid changes in the local odor concentrations, and instantaneous odor gradients often do not point toward the source (*Celani et al., 2014*; *Crimaldi and Koseff, 2001*). Encounters between the animal and odorized packets of air are intermittent, with durations and frequencies spanning many orders of magnitude (*Celani et al., 2014*). Moreover, distinct flow conditions result in distinct spatio-temporal statistics: near boundaries and with lower mean wind speeds, odor plumes are smoother, with odor concentrations consistently above detectable thresholds (*Connor et al., 2018*). But rough-ness in the physical landscape – sands, rough terrain, vegetation – and shifting winds can cause plumes

to break up into discrete odor filaments, interspersed with long periods of undetectable concentrations (*Cardé and Willis, 2008*; *Murlis et al., 1992*; *Riffell et al., 2008*). There, encounters with odor filaments can occur over a wide range of frequencies from 0.1 Hz (*Riffell et al., 2008*) to 5 Hz or more (*Demir et al., 2020*).

To navigate plumes exhibiting this degree of temporal complexity, animals must be able to detect odor encounters quickly and accurately. Indeed, many organisms have evolved olfactory receptor neurons (ORNs) that respond to chemical signals with high temporal precision (*Gorur-Shandilya et al., 2017*; *Jacob et al., 2017*; *Nagel and Wilson, 2011*; *Szyszka et al., 2014*; *Szyszka et al., 2012*). ORN firing responses are strongly time-locked to the arrival time of an odor (*Gorur-Shandilya et al., 2017*), and fast synaptic mechanisms (*Fox and Nagel, 2021*; *Martelli et al., 2013*) allow this information to be passed quickly downstream, within milliseconds, to projection neurons (PNs) in the antennal lobe, driving rapid behavioral responses (*Bhandawat et al., 2010*). Such precision has been suggested to allow accurate encoding of temporal features of the odor signal (*Nagel et al., 2015*), such as the frequency of odor arrivals.

In addition to these fast responses, *Drosophila* ORNs also adapt their firing rates and gain to prolonged stimuli (*Cao et al., 2016*; *Gorur-Shandilya et al., 2017*; *Nagel and Wilson, 2011*), priming them to accurately encode future odor signals (*Kadakia and Emonet, 2019*) without losing temporal precision as intensity changes (*Gorur-Shandilya et al., 2017*; *Martelli et al., 2013*). Likewise, in honeybees, the temporal resolution of odor pulses increases over time in a pulsed odor environment (*Szyszka et al., 2014*), while in the moth *Agrotis ipsilon*, ORN responses adjust to optimally encode odor signals that occur most frequently in the environment (*Levakova et al., 2018*). Olfactory neurons in insects are thus sensitive to the temporal features of odor signals on both short and long timescales (*Nagel et al., 2015*).

Temporal precision in olfaction extends beyond insects. In mice, plume dynamics as fast as tens of milliseconds are encoded downstream in mitral and tufted cells (*Ackels et al., 2021*). In crustaceans, odors are encoded by bursting ORNs (or bORNs), which burst only if odors arrive at some phase relative to an intrinsic bursting cycle (*Park et al., 2014*). These cycles vary over orders of magnitudes across the bORN population, providing a natural template to encode the timing between odor arrivals (*Park et al., 2016*).

Naturally, such precisely resolved temporal odor information shapes navigational decisions. When tracking pheromones, flying male moths fly faster and straighter upwind when receiving odor hits at higher frequencies than lower ones (*Mafra-Neto and Cardé, 1994*; *Vickers and Baker, 1994*). Walking silkworm moths switch from zigzagging motion to straighter trajectories upwind in higher-frequency environments (*Kanzaki et al., 1992*). One model (*Vickers and Baker, 1994*) has suggested that odor hits suppress an otherwise persistent internal counterturning mechanism, allowing moths to maintain straight trajectories if odors are frequent or long. Alternatively, flying flies counterturn shortly after passing through the odor (*Budick and Dickinson, 2006*; *van Breugel and Dickinson, 2014*), indicating that counterturning can be also driven by the loss of the plume rather than an internal mechanism. In water, crabs navigate successfully in environments with higher-odor intermittency, but fail to find odor sources as pulses become more infrequent (*Keller and Weissburg, 2004*).

Two recent studies in *eLife* have quantified in great detail, using both experiment and extensive mathematical modeling, the olfactory navigational strategies of walking *Drosophila* in wind tunnels. One of these (*Álvarez-Salvado et al., 2018*) focused on spatially uniform but temporally varying environments, where the odor was presented in spatially uniform pulses lasting anywhere from 1 to 10 s. In this environment, walking flies maintained upwind headings and increased walking speed over the duration of the odor pulses, albeit with a degree of desensitization over time. This suggests that when odor encounters are long and persistent the *intermittency* of the odor signal – which we define to be the percentage of time the odor signal is above threshold – is a main driver of navigational decisions. The second study (*Demir et al., 2020*) instead challenged flies to navigate spatiotemporally complex odor plumes that were generated by stochastically perturbing a thin ribbon of odor. In this plume, odor encounters were much shorter (~0.1–0.3 s), more frequent (~3 Hz), and less predictable. In that study, fly navigation was reproduced by a model in which only the *frequency* of odor encounters controlled upwind orientation, independent of their duration or concentration. These two studies used the same organism with the same

locomotive repertoire. The two distinct models they uncovered naively suggest that flies are able to sense distinct temporal features of odor plumes and use these various inputs to shape navigational decisions.

Here, we use mathematical modeling and numerical simulations to investigate how and under what conditions these two temporal features – odor intermittency and encounter frequency – can enhance the navigation of turbulent odor plumes. To examine the contribution to navigation from these two temporal features alone, we ignore other sensory modalities, such as concentration gradient sensing, bilateral sensing, and vision. We first demonstrate analytically that the dynamical model proposed in the first study above picks out (in appropriate limits) odor signal intermittency, while the model in the second study responds to the frequency of odor hits. These two temporal features are complementary and can be varied independently, forming a natural basis of temporal sensing. We devised a simple model that incorporates intermittency sensing and frequency sensing in a minimal way, and uses these two 'sensors' to drive upwind orientations. Using agent-based simulations, we first show that this combined model requires both sensors to successfully navigate both measured plumes used in the two studies. We then applied the navigational model to simulated plumes, leveraging an advecting-diffusing packet framework that mimics odor motion in turbulent flows (*Farrell et al., 2002*). We find that to robustly navigate a variety of plumes agents should use both intermittency and frequency sensing. However, there is a trade-off in performance when using both temporal features simultaneously, which persists across a variety of plumes. This predicts a strong benefit to modulating the weight of these two sensors, and we propose simple experiments to test whether flies or other insects indeed carry out such adaptation on slower timescales. Finally, we explore how simultaneous frequency and intermittency sensing is enabled by the *Drosophila* olfactory circuit, using previously developed models of ORNs and their synaptic connections to PNs (*Gorur-Shandilya et al., 2017*; *Nagel et al., 2015*). We find that PNs respond independently to both features and enable effective navigation through various environments, suggesting that the first stage of olfactory processing is appropriately tuned for naturalistic navigation tasks.

## Results

### Two experimentally constrained models implicate distinct odor signal features in olfactory navigation

Our study is motivated by two models recently extracted from experimental observations of walking *Drosophila* navigating odor plumes (*Álvarez-Salvado et al., 2018*; *Demir et al., 2020*). Here, we examine how they each respond to distinct temporal features of the odor concentration. We focus on temporal changes in odor concentration rather than odor flux (which depends also on air speed) as *Drosophila melanogaster* ORN responses are invariant to air speed (*Zhou and Wilson, 2012*). In the first model (*Figure 1A*; *Álvarez-Salvado et al., 2018*), the instantaneous odor concentration $odor(t)$ is first compressed into the range 0–1 using an adaptive Hill function:

$$C(t) = \frac{odor(t)}{odor(t) + k_d + A(t)}. \tag{1}$$

The half-max is set by $A(t)$, a low-pass-filtered sliding average of the instantaneous odor concentration

$$\tau_A \frac{dA}{dt} = odor(t) - A(t). \tag{2}$$

This mimics the gain adaptation of ORNs to the mean signal (*Cao et al., 2016*; *Gorur-Shandilya et al., 2017*). At the onset of a sudden increase in odor concentration, the compressed signal $C(t)$ increases instantaneously before relaxing back to ~0.5 with timescale $\tau_A = 9.8$ s. The compressed signal $C(t)$ is then exponentially filtered into an 'ON' function,

$$ON(t) = \int_0^t \frac{1}{\tau_{ON}} \cdot e^{\frac{t'-t}{\tau_{ON}}} \cdot C(t') \, dt', \tag{3}$$

which drives odor-elicited behavioral actions. When $ON(t)$ is high, the fly accelerates and biases its heading upwind; when $ON(t)$ is low, the fly's orientation randomizes and drifts downwind and its

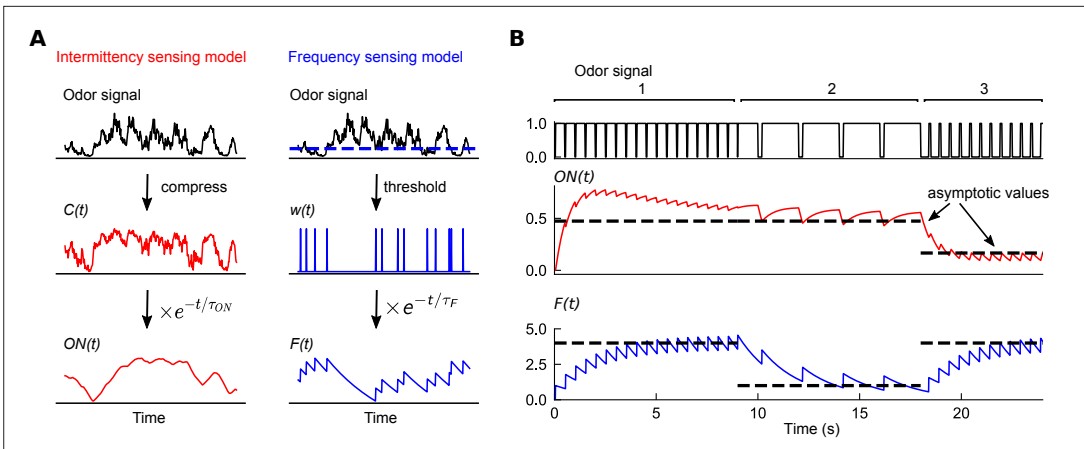

**Figure 1.** Filters extracted from experiment capture distinct temporal features of odor signals. (**A**) Two experimentally informed models (**Álvarez-Salvado et al., 2018**; **Demir et al., 2020**) of *Drosophila* olfactory navigation transform odor signals in distinct ways. Left column: the intermittency model compresses the odor signal with an adaptive nonlinearity into a representation $C(t)$, bounded between 0 and 1. $C(t)$ is then exponentially filtered with timescale $\tau_{ON} = 0.72s$ to generate $ON(t)$. Right column: the frequency model thresholds the odor signal (dashed line in top plot) into a binary representation $w(t)$, which is then passed through an exponential filter with timescale $\tau_F = 2s$ to generate $F(t)$. (**B**) Response of each of the models (bottom two plots) to a binary odor signal (top plot) of high intermittency, high frequency (region 1), high intermittency, low frequency (region 2), and low intermittency, high frequency (region 3). The intermittency model is sensitive to the intermittency of the signal – in regions 1 and 2, it approaches a high value asymptotically, but a low value when intermittency is low, even if the frequency remains high (region 3). The asymptotic values of the intermittency model (dashed lines) are $\frac{I}{1+I}$, where *I* is signal intermittency (Materials and methods). Conversely, the frequency model exhibits sensitivity to the frequency of encounters, tending asymptotically towards $f \cdot \tau_F$, where *f* is the signal frequency (dashed line). The frequencies in the three regions are 2 Hz, 0.5 Hz, and 2 Hz, the encounter durations are 0.45 s, 1.8 s, and 0.1 s, and the intermittencies are thus 0.9, 0.9, and 0.1.

walking speed reduces (**Álvarez-Salvado et al., 2018**). We show analytically (Materials and methods) that the value of $ON(t)$ – and therefore the navigational actions – is largely determined by the intermittency of the odor signal, defined as the percentage of time an odor signal is present. Thus, we refer to this model as the *intermittency model*.

In the second model (**Figure 1A**; **Demir et al., 2020**), a detection threshold is used to detect when the odor arrives. This results in a binary time series $w(t)$, which spikes as a $\delta$-function each time the odor concentration crosses the threshold from below, and is 0 otherwise. The frequency of odor encounters is then estimated by filtering $w(t)$ with an exponential:

$$F(t) = \int_0^t e^{\frac{t'-t}{\tau_F}} w(t')\, dt'. \tag{4}$$

Thus, $F(t)$ rises by 1 at each threshold crossing, before decaying exponentially with timescale $t_w$ until the next odor hit. In this model, $F(t)$ plays a similar role as $ON(t)$ in the previous model, in that it drives behavioral response to odors. When $F(t)$ increases, flies increase their bias upwind and stop less frequently and for a shorter time (**Demir et al., 2020**). Since $F(t)$ is effectively a running average of the frequency of odor hits, we refer to this model as the *frequency model*.

To illustrate how each of these two sensory modalities respond to the temporal features of odor signals, we plotted the output of each filter in response to square-wave odor pulses of given frequency and intermittency (**Figure 1B**). These two features can be independently tuned – an odor signal can be high frequency and high intermittency if the whiffs (periods above threshold) are interrupted frequently with blank periods that are very short (region 1 in **Figure 1B**), while it can have high intermittency but low whiff frequency if whiffs are interrupted with short blank periods occurring more sparsely (region 2 in **Figure 1B**). In the first two regions of the signal, where intermittency is high, the response of the $ON(t)$ model approaches a high value after an initial transient, while it drops to a lower steady state in region 3 where the signal intermittency is lower. The steady-state response of $ON(t)$ is sensitive to the

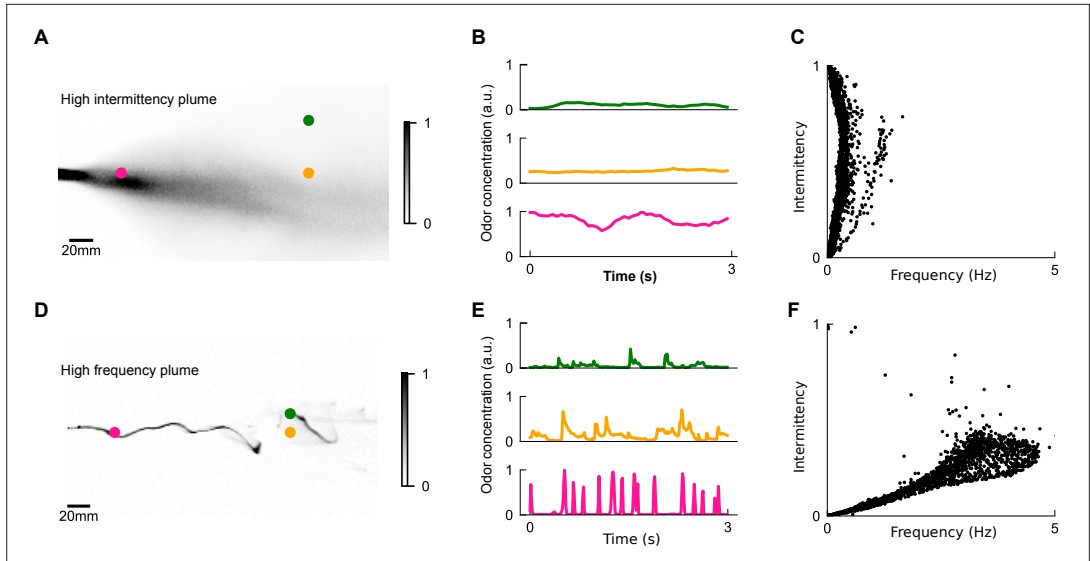

**Figure 2.** The differing temporal statistics of odor plumes. (**A**) Snapshot of measured high-intermittency plume, reproduced from data in *Connor et al., 2018*. Colored dots: locations corresponding to odor series in (**B**). (**B**) Odor concentration time series at different locations in high-intermittency plume. (**C**) Intermittency versus whiff frequency for 10,000 uniformly distributed points in the high-intermittency plume. Statistics were calculated over the length of the full video. We see a range of intermittencies and many points with high intermittencies but relatively low frequencies. (**D, E**) High-frequency plume and representative time series, reproduced from data in *Demir et al., 2020*. (**F**) Analogous to (**C**) for the high-frequency plume. Data is clustered within a higher range of frequencies but low intermittencies.

signal intermittency, but is independent of the whiff frequency, as indicated by the average response asymptote $\frac{I}{1+I}$, which monotonically increases with intermittency (Materials and methods). In contrast, the frequency model responds strongly in regions 1 and 3, where whiff frequency is high, consistent with its asymptotic response $f \cdot \tau_F$ (Materials and methods). This happens irrespective of the disparity in signal intermittency between these regions (*Figure 1B*, bottom trace). Note that both models are sensitive to the temporal characteristics of the signal, but not absolute concentration.

Though these two models were extracted from the same model organism with the same locomotive repertoire – fruit files walking in a 2D arena – the experiments were performed in very different odor and flow conditions. The intermittency model was first extracted from flies navigating a uniformly odorized region of odor within a laminar airflow (*Álvarez-Salvado et al., 2018*). Using simulations, the model was then shown to qualitatively recapitulate navigational behavior in a measured near-bed turbulent plume (*Connor et al., 2018*; *Figure 2A*), which we call the high-intermittency plume, in which the odor signal was ever-present and varied on relatively long timescales of several seconds or more (*Figure 2B*). In contrast, the frequency model was fit to trajectories of flies navigating a plume with a high degree of spatial complexity (*Figure 2D*) generated by perturbing a fast laminar flow with stochastic lateral jets, which we call the high-frequency plume. In that experiment, odor whiffs occurred frequently (2–5 Hz) (*Figure 2E and F*) and were much shorter (~100 ms) (*Figure 2E*). The two navigational models these experiments informed were clearly shaped by the plumes' natural features: in the first, odor intermittency reached as high as 100% and whiff frequencies rarely surpassed 1 Hz (*Figure 2C*), whereas in the latter, the signal had intermittency mostly below 30% but whiff frequencies of several Hz (*Figure 2F*). Together, these two experiments and corresponding models suggest that flies use both odor frequency and intermittency to navigate upwind in different environments. This prompted us to ask how this dual-sensing capability might enhance the efficacy and robustness of navigation in different conditions.

## Dual intermittency and frequency sensing enhances navigation robustness in distinct environments

To next investigate how these dual-sensing capabilities – odor intermittency sensing and frequency sensing – shape navigational performance in distinct odor landscapes, we incorporated them into a combined navigational model. It is known that odor signals influence many behavioral actions, including accelerating, turning, and stopping (*Álvarez-Salvado et al., 2018*; *Baker and Vickers, 1997*; *Demir et al., 2020*; *Mafra-Neto and Cardé, 1994*; *Vickers and Baker, 1994*). Given the near-universal response of insects to turn upwind or bias their turns upwind in the presence of odor (*Baker et al., 2018*), here we assumed agents walk at a constant speed unless they are turning and focused on signal-driven changes in orientation. Turns occur randomly at a Poisson rate $\lambda_{turn}$, and turn magnitudes are sampled from a normal distribution $N\left(30^o, 8^o\right)$ as found before (*Demir et al., 2020*). Turn directions (sign of the orientation change) are modeled as

$$p\left(\text{turn upwind}\middle|\text{turning}\right) = \frac{1}{1+e^{-g_I ON - g_F F}}. \tag{5}$$

Thus, the likelihood that a turn is directed upwind (versus downwind) increases sigmoidally with a linear combination of $F\left(t\right)$ and $ON\left(t\right)$. In the absence of signal, upwind and downwind turns are equally likely: $P\left(\text{upwind|turn}\right) = 0.5$. To allow frequency sensing to be adaptive, we set the detection threshold for $F\left(t\right)$ to be variable and equal to $\frac{1}{2}A\left(t\right)$, where $A\left(t\right)$ is defined in *Equation 2*. The 'sensor gains' $g_I$ and $g_F$ were set to 3.9 and 0.2, respectively, by comparing to experimental data (Materials and methods). For now, we hold the gains fixed at these 'base' values $g_{I0} = 3.9$ and $g_{F0} = 0.2$; below, we investigate the performance of different $g_I$ and $g_F$. Finally, we define intermittency-only and frequency-only sensing models by setting $g_F$ and $g_I$ to 0, respectively.

To examine how frequency and intermittency contribute to navigational performance in this combined model, we simulated $N$ agents navigating both the high-intermittency and high-frequency plumes. The initial position and orientations of the agents were randomized uniformly. Performance was quantified as the fraction of agents that reach within 15 mm of the source in the presence of an odor signal, $\frac{N_s}{N}$, minus the fraction of agents, $\frac{N_c}{N}$, that reach the source by chance, that is, when no signal is present. Individual trajectories of successful flies in either plume look similar: when oriented away from the source, agents are quickly able to reorient within the plume region and navigate to the source with relatively straight trajectories combined with occasional corrective kinks (*Figure 3B*). Overall, agents navigated successfully in both plumes (*Figure 3C*), and performance was relatively robust to initial angle and position (*Figure 3D*). However, when either frequency sensing $g_F = 0$ or intermittency sensing $g_I = 0$ was removed, performance degraded (*Figure 3D*) in one of the plumes and became more sensitive to initial conditions. Though not wholly surprising that removing sensors degrades performance, this suggests that a simple linear combination robustly navigates two disparate odor plumes, without exhibiting any obvious failure modes due to interference between sensors.

Our upwind bias function (*Equation 5*), though phenomenological, is a natural choice in that it allows an increased upwind response to both the $ON$ and $F$ filters. In fact, it very closely approximates a logical OR gate for the two filters (Materials and methods; *Equations 34-35*). This raises the question of whether this particular logical operation is ideal. We similarly investigated an AND gate implementation, finding clear failure modes (Materials and methods).

We expect that the two sensors do not contribute equally at all times to the navigation and that the relative contribution of either sensor may depend on plume statistics or on the location within a plume (*Rigolli et al., 2021*). For example, in the high-frequency plume, the intermittency sensor is likely to also be active near the plume centerline, where the signal is more likely to be present, while in the high-intermittency plume the frequency sensor is likely to be active on the edges where the presence of odor is less certain. To quantify this, we measured the relative weight of each sensor $\frac{g_I ON(t) - g_F F(t)}{g_I ON(t) + g_F F(t)}$, which interpolates between pure intermittency sensing (+1) and pure frequency sensing (–1). As expected, the intermittency sensor dominates in the high-intermittency plume, whereas the frequency sensor dominates in the high-frequency plume (*Figure 3E*). Still, this dominance is not absolute. For example, frequency sensing plays a role near the conical boundary of the high-intermittency plume. Likewise, intermittency contributes along the centerline of the high-frequency plume.

These modest but significant contributions led us to next wonder how the sensors might be relatively weighted to optimize navigational performance and how this weighting might change in different

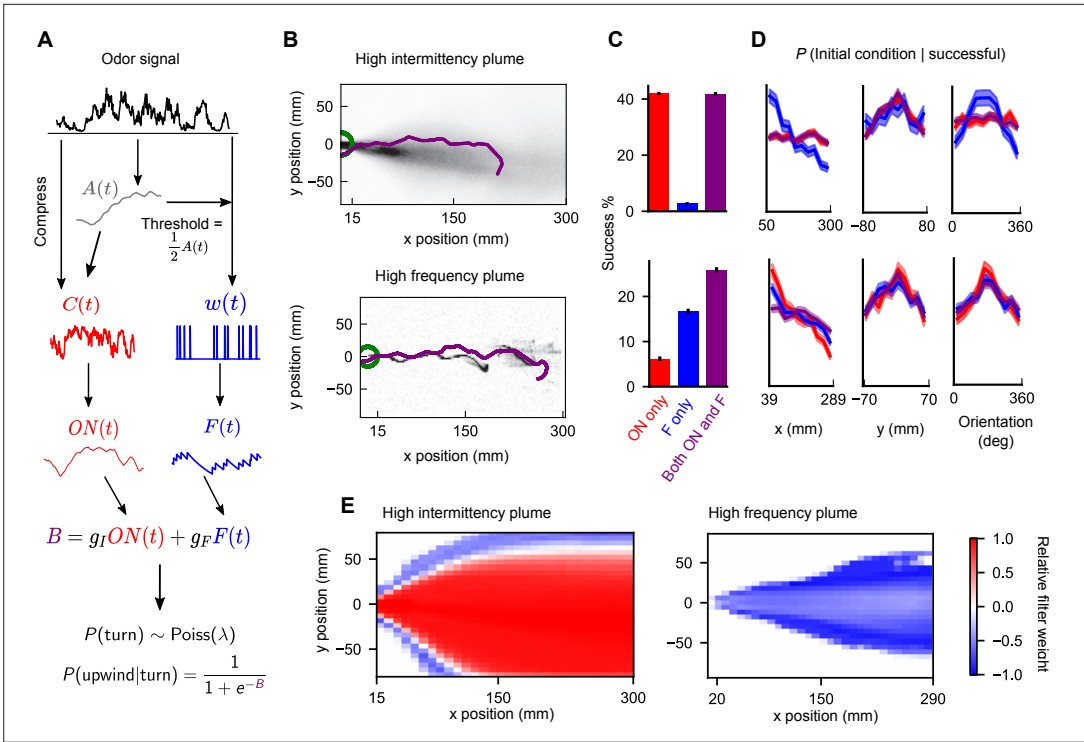

**Figure 3.** Sensing both intermittency and frequency enables navigation across diverse plumes. (**A**) Our model linearly combines an intermittency sensor (red) and whiff frequency sensor (blue) to bias upwind motion. For both sensors, the odor signal is transformed using an adaptive compression step $A(t)$ (*Álvarez-Salvado et al., 2018*) before being converted into a turning bias. Following (*Demir et al., 2020*), turns occur stochastically at a constant Poisson rate $\lambda_{turn}$, while the sensor output $B$ biases the likelihood that turns are upwind. Turn magnitudes are chosen from a normal distribution with mean 30° and SD 8° (*Demir et al., 2020*). (**B**) Example successful trajectories in the high-intermittency and high-frequency plume (*Figure 2*). (**C**) Percentage of agents that reach within 15 mm of the source when signal is present minus same percentage when signal is absent, for the model with only intermittency sensing ($g_F = 0$; red), only frequency sensing ($g_I = 0$; blue), or both ($g_F, g_I$ nonzero; purple), in the high-intermittency plume (top) and high-frequency plume (bottom). Error bars: SEM calculated by bootstrapping the data 1000 times (Materials and methods). (**D**) Distribution of initial downwind position $x$ (first column), crosswind position $y$ (second column), and orientation (third column) for successful agents for the high-intermittency (top row) and high-frequency (bottom row) plumes. Colors correspond to same models as in (**C**). Upwind heading is 180°, and shaded regions represent SEMs obtained from bootstrapping (Materials and methods) (**E**) Time-averaged relative filter weight $\frac{g_I ON - g_F F}{g_I ON + g_F F}$ for different points in the two plumes.

The online version of this article includes the following figure supplement(s) for figure 3:

**Figure supplement 1.** Fixed-threshold navigation model produces results similar to navigation model with adaptive intensity compression.

plumes. Therefore, for tractability, we constructed a simpler model that eliminated some parameters. Firstly, we retained the frequency sensor $F(t)$ (*Equation 4*), but used a fixed odor detection threshold $K$ rather than an adaptive threshold as before. Secondly, we replaced the $ON(t)$ function with:

$$I(t) = \frac{1}{2} \cdot \int_0^t \frac{1}{\tau_I} \cdot e^{\frac{t'-t}{\tau_I}} \cdot \Theta\left(\text{odor}\left(t'\right) - K\right) dt'. \tag{6}$$

where $\Theta$ is the Heaviside step function. The primary change from $ON(t)$ is the replacement of adaptive odor compression with a fixed binarizing odor threshold. The factor of ½ is kept for ease of comparison between $I(t)$ and $ON(t)$, so that both filters asymptotically approach ½ in the presence of continuous odor (Materials and methods). Filtering timescales were set at $\tau_I = \tau_F = 2s$ for both $I(t)$ and $F(t)$. While these changes do affect some quantities, like the relative filter weight in the two environments, the overall effect on navigational success is minimal (*Figure 3—figure supplement 1*).

Thus, to study the effect of various model parameters in detail, we used this simplified model for all further investigations.

## Optimal performance requires distinct weighting of frequency and intermittency in different environments

Upwind bias, and therefore navigation performance, depends on the sensor gains (*Equation 5*), which up to now we have fixed to experimentally informed values (the 'base' gains). To investigate the influence of relative sensor weight in navigation, we quantified navigational performance as a function of both the sensor weights $g_I$ and $g_F$ and the plume's spatiotemporal complexity. To remove constraints due to the limited spatial and temporal resolution of the recorded plume videos, and to easily investigate a wide range of environments, we switched to simulated plumes using a simple dispersion model (*Farrell et al., 2002*). Gaussian packets of odor are released from a source at a fixed Poisson rate $\lambda$ and advected by a velocity field composed of a uniform downwind velocity $U$. Normally distributed random perturbations $\eta_x$ and $\eta_y$ are added to the packet positions in the crosswind and downwind directions, respectively, at each time step, to account for the effects of turbulent diffusivity. The turbulent diffusivity models the effects of turbulent eddies as a diffusive process, but with diffusion constant $\kappa$ that can greatly exceed molecular diffusivity. In addition, the Gaussian packets grow in size with an effective diffusivity $D$ to account for the combined effects of molecular diffusion and smaller eddies in the wind flow (*Figure 4A and B*). Varying $U$ and $D$ allowed us to generate plumes with diverse temporal statistics. $U = 36\,\text{mm/s}$ and $D = 52\,\text{mm}^2/\text{s}$ resulted in a plume with longer whiff durations and high intermittency (*Figure 4C and E*). Increasing the wind speed to $U = 300$ mm/s and decreasing effective diffusivity to $D = 10\,\text{mm}^2/\text{s}$ resulted instead in a high-frequency plume with much shorter whiffs (*Figure 4D and F*). In each plume, we simulated 10,000 agents with uniformly distributed initial position and heading angle, where each agent navigated with a fixed set of gains $g_I$ and $g_F$. We investigated various choices of $g_I$ and $g_F$, from 0 to 50× the base gains.

The $g_I$ and $g_F$ maximizing performance in our simulated high-intermittency plume was reasonably constrained, with a clear maximum occurring around the experimentally derived base gain (*Figure 4—figure supplement 1*). However, in the simulated high-frequency plume, a variety of gains led to similarly maximal performance (*Figure 4—figure supplement 1*), including some with values an order of magnitude larger than the base gains. Performance was largely independent of $\tau_I$ over nearly two orders of magnitude (unchanged even for a null algorithm that drives upwind orientation whenever odor is present, i.e., $\tau_I = 0$) and scaled with $\tau_F$ in a way that could be absorbed into the $g_F$, (*Figure 4—figure supplement 2*; Materials and methods), so these trends were fundamentally due to the sensor gains rather than other model features. On the other hand, models with extreme gain factors could compound the effects of noise, leading to a lack of robustness in natural conditions. We therefore added Gaussian noise to the *I* and *F* filters – noise amplitude was 5% of the average value of *I* (F) in the center of the simulated high-intermittency (high-frequency) plume. This removed maxima at high gains but retained clear maxima at lower gains (*Figure 4G and H*). Interestingly, the unique maxima sat fairly close to the base gain values (values of 1 in *Figure 4G and H*), suggesting a degree of tuning within the biological fly olfactory circuit. Finally, the optimal gains for the simulated high-intermittency and high-frequency plumes had $g_F = 0$ and $g_I = 0$, respectively, indicating that optimal performance in either plume requires silencing the nonrelevant sensors. This inherent trade-off illustrates that simply augmenting the sensory capability can at times degrade performance. This suggests a benefit for sensor specialization in distinct environments.

## Performance trade-off between intermittency sensing and frequency sensing in different environments

To get a better understanding of how navigational performance in these two simulated plumes depends on the sensor weights, we did a tighter sweep of gains near the performance maxima (*Figure 4G and H*) for each plume. For each set of gains, we then plotted performance in the high-intermittency plume against that in the high-frequency plume. For comparison, we also plotted the set of gains $(g_I^*, g_F^*)$ that maximized the geometric mean of normalized success in both plumes (indicated in *Figure 4I*). The resulting scatterplot quantifies the performance in the two plumes for different navigational models, where each model is parameterized by its sensor weights $g_I$ and $g_F$. In general, the scatterplot fills out a region near the origin, bounded by a curve that forms a 'Pareto front' of

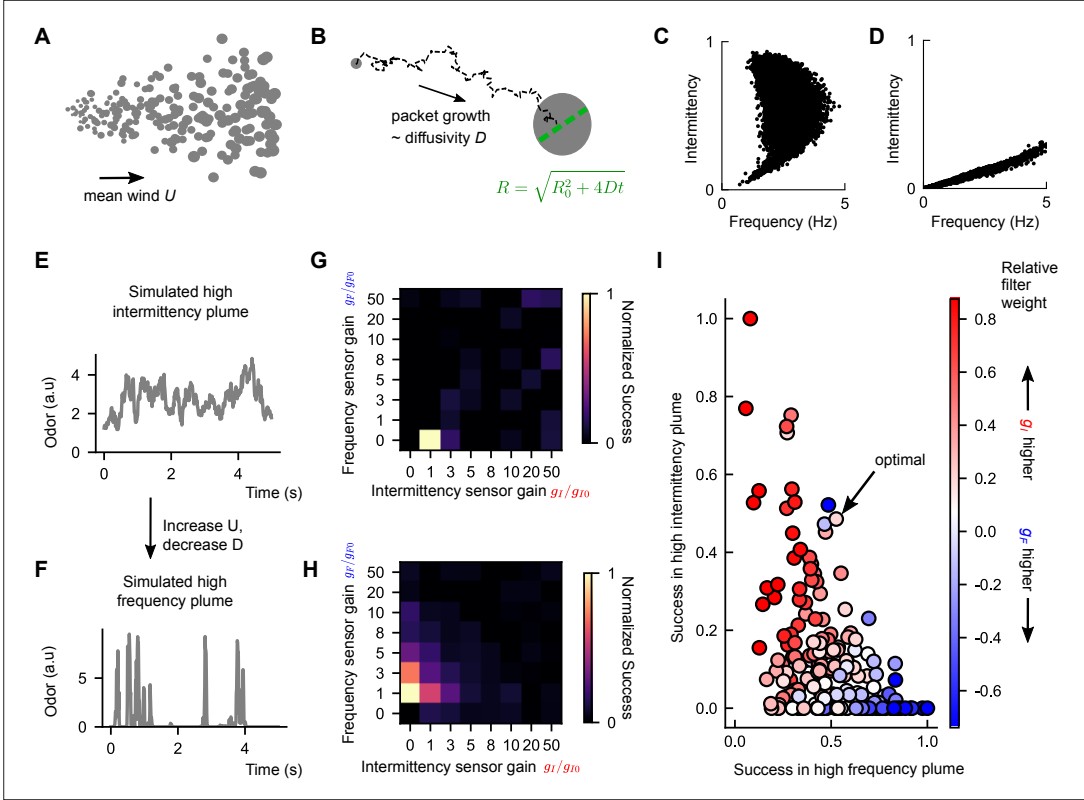

**Figure 4.** Performance trade-off between intermittency and frequency sensing in two diverse turbulent plumes. (**A**) Example of a simulated odor plume, following the framework in *Farrell et al., 2002*. Gray circles denote Gaussian odor packets. (**B**) Example trajectory of a single-odor packet in these simulations and illustration of its growth. (**C**) Same as *Figure 2C* but for the simulated high-intermittency plume. (**D**) Same as (**C**) but for the simulated high-frequency plume. (**E**) Example odor concentration time series in a simulated high-intermittency plume. (**F**) Same as (**C**), for a high-frequency plume. (**G**) Normalized success percentage $S$ within the simulated high-intermittency plume after adding noise to $I$ and $F$. $S$ is computed by first calculating the success percentage as in *Figure 3C* for each pair of gains $(g_I, g_F)$ and then normalizing by the maximum success percentage over all $(g_I, g_F)$. Gains are measured in multiples of the base gains, defined in Materials and methods. (**H**) Same as (**G**), but for the simulated high-frequency plume. (**I**) $S$ in the simulated high-intermittency plume versus $S$ in the simulated high-frequency plume, where each dot represents a different $(g_I, g_F)$. Points are colored by the relative weighting of the two sensors (see Materials and methods for calculation details). Note here that a finer set of gains was considered than in (**G**) and (**H**) and normalization was done with respect to these gains. The pair $(g_I, g_F)$ that maximized the geometric mean of normalized success percentage across the two plumes is indicated as optimal. The concavity of the front suggests a sharp trade-off in performance in one plume versus the other.

The online version of this article includes the following figure supplement(s) for figure 4:

**Figure supplement 1.** Performance of different sets of gains without filter noise.

**Figure supplement 2.** The effect of changing filter timescales on navigation success.

navigational performance. This Pareto front reveals a performance trade-off for the different models: combinations of $g_I$ and $g_F$ that are weighted toward $I$ do better in the high-intermittency plume, while combinations weighted toward $F$ outperform in the high-frequency plume (*Figure 4I*). There was no fixed set of gains that performs optimally in both plumes. Importantly, the apparent concavity of the Pareto front illustrates a somewhat steep trade-off and suggests that flies might be better off modulating gains and switching between using intermittency and frequency sensors to bias upwind motion, as opposed to using both simultaneously.

We then wondered how this trade-off manifests across a more diverse spectrum of plumes. The computational simplicity of the turbulent plume model allowed us to study a wide array of turbulent plumes differing in their temporal statistics. We fixed the gains to the values that optimized the geometric mean between the high-intermittency and high-frequency plumes, $(g_I^*, g_F^*)$, and then varied

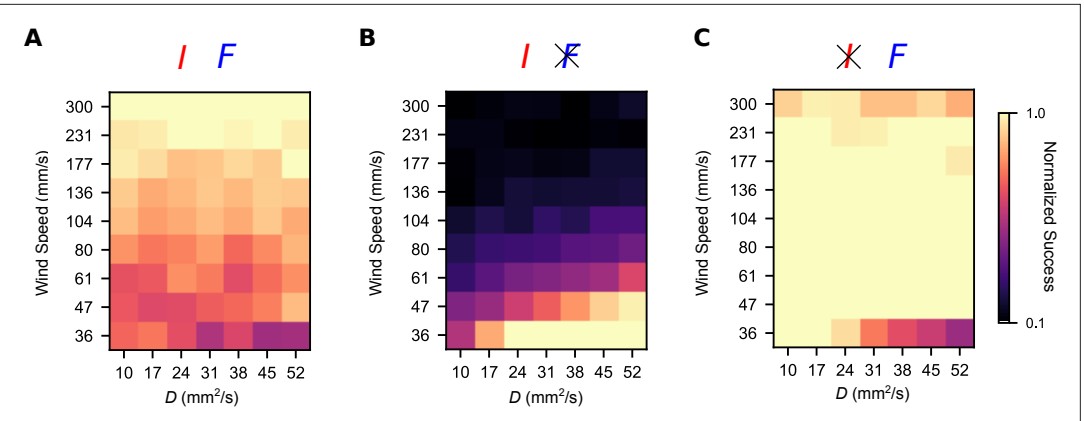

**Figure 5.** Simultaneous intermittency and frequency sensing maintains steady performance across a spectrum of odor environments, but does not allow for optimal performance. Normalized success percentage for a frequency and intermittency-sensing model (**A**), only intermittency-sensing model (**B**), and only frequency-sensing model (**C**) for a range of simulated odor plumes. Success percentage is normalized such that the best performance of the three models is set to 1 for each environment. Gains for (**A**) were chosen to optimize the geometric mean of performance in the simulated high-intermittency and high-frequency plumes. Gains in (**B**) and (**C**) were chosen by taking the gains in (**A**) and then setting $g_F$ (**A**) and $g_I$ (**C**) to 0.

the environmental parameters $U$ and $D$ to smoothly interpolate between the high-frequency and high-intermittency plumes investigated above. Success was roughly uniform in the different environments (*Figure 5A*). However, removing the frequency sensor ($g_F = 0$) significantly improved performance in the slowly advecting and highly diffusive plumes (low $U$; high $D$), which tend to be smoother in their concentration profiles. The reverse was true when we removed intermittency sensing ($g_I = 0$), exemplifying a trade-off in navigational performance that persists across this wide range of odor environments. Together with the results presented above (*Figure 3*), this suggests that while a naïve summation of temporal sensors may be beneficial in some cases, in general, navigation can always be improved by some degree of specialization.

## Biophysical neural filtering of odor signals enables independent frequency and intermittency sensing and aids in navigation

Our results so far suggest that dual sensing of two complementary odor signal features, intermittency and frequency, aids navigation across a diversity of odor plumes, albeit with a trade-off. To what extent is this dual-sensing capability enabled by the *Drosophila* olfactory circuit? Prior experimental and modeling work has shown that synaptic and circuit mechanisms in the olfactory periphery allow for accurate signal transmission across a range of frequencies (*Martelli and Fiala, 2019*; *Nagel et al., 2015*), while fast ORN adaptation allows signals to be encoded without saturation (*Gorur-Shandilya et al., 2017*). These various mechanisms suggest that the natural structure of the fly olfactory circuit may be well-primed for robust encoding of multiple temporal features of the odor signal.

We thus combined prior models (*Gorur-Shandilya et al., 2017*; *Nagel et al., 2015*) into a single model of odor binding, ORN firing, and PN response, and fed this naively into a behavioral module to investigate navigational performance. At the first stage of processing, odors bind an olfactory receptor/co-receptor (Or/Orco) complex, which can be active (ion channel open) or inactive (closed). Assuming fast binding dynamics, the average activity $a$ of the complex is

$$a = \left(1 + e^\epsilon \cdot \frac{1 + \frac{C}{K_{off}}}{1 + \frac{C}{K_{on}}}\right)^{-1} \tag{7}$$

where $C$ is the odor concentration, $\epsilon$ is the free energy difference between the active and inactive states when unbound, and where the dissociation constant between odorant and the complex in the inactive state, $K_{off}$, is much higher than that for the active state, $K_{on}$. To model adaptation, receptor activity feeds back into $\epsilon$ via

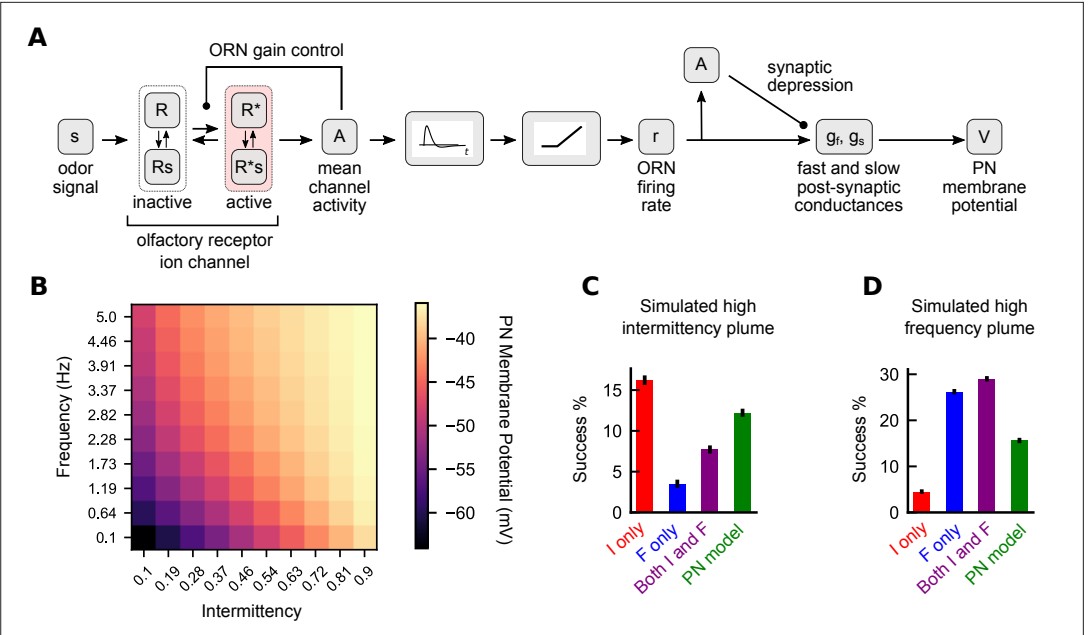

**Figure 6.** A biophysical signal transduction model allows for simultaneous frequency and intermittency sensing and performs similarly to a combined model. (**A**) A schematic for how we combine the models of *Gorur-Shandilya et al., 2017* and *Nagel et al., 2015* to convert odor signals to projection neuron (PN) membrane potentials. (**B**) Time-averaged PN membrane potentials in square-wave environments of different frequency and intermittency. Responses were simulated for 30 s and last 20 s were averaged. (**C**) Performance of different navigation models considered in the simulated high-intermittency plume. Success was computed as in *Figures 3 and 4*. (**D**) Same as (**C**) but for the simulated high-frequency plume. Note that in (**C**) and (**D**) no noise was added to the filter outputs for any of the models.

$$\frac{d\epsilon}{dt} = \beta\left(a - a_0\right) \tag{8}$$

where $\beta$ is an adaptation rate and $a_0$ is a baseline activity. ORN firing rate is then obtained by passing $a$ through a linear filter and static nonlinearity (*Figure 6A*; see Materials and methods). Finally, $\epsilon$ is bounded from below ($\epsilon > \epsilon_L$) so that ORNs shut off with sufficiently weak odor.

ORN firing rate is converted into a PN membrane potential through a postsynaptic conductance with two timescales (*Nagel et al., 2015*). Conductances are weakened over time via synaptic depression, also with two timescales (*Figure 6A*). This depression is modeled by a scaling factor of the conductance, $A_{fast}\left(t\right)$ (analogously for $A_{slow}\left(t\right)$):

$$\frac{dA_{fast}}{dt} = -r_{fast}s\left(t\right)A_{fast}\left(t\right) + \frac{1 - A_{fast}\left(t\right)}{\tau_{Afast}} \tag{9}$$

where $s$ is the ORN firing rate, $r_{fast}$ is the rate that $A$ decays with increased firing rate, and $\tau_{Afast}$ is the timescale it takes for $A_{fast}$ to relax back to 1. This scaling factor then affects the synaptic conductance via

$$\frac{dq_{fast}}{dt} = k_{fast}s\left(t\right)\cdot A_{fast}\left(t\right) - \frac{q_{fast}\left(t\right)}{\tau_{gfast}} \tag{10}$$

where $q_{fast}$ is the fast conductance (analogous for the slow conductance). The fast and slow conductances are summed to give a total synaptic conductance $q_{syn}$. The PN membrane potential $V\left(t\right)$ then obeys

$$\frac{dV}{dt} = \frac{-V\left(t\right) - E_{leak} + q_{syn}\left(t\right)R_m\left(V\left(t\right) - E_{syn}\right)}{\tau_m} \tag{11}$$

where $E_{leak}, E_{syn}$ are the reversal potential for leak and synaptic currents, respectively, $R_m$ is the resistance of the membrane, and $\tau_m$ is the timescale of the membrane. For parameter values, see Materials and methods and **Nagel et al., 2015**.

We first looked to see how the PN membrane potential responds to environments of different temporal statistics. As in **Figure 1**, we simulated the potential in square-wave environments of varying frequencies and intermittencies. We find that average membrane potential increases with frequency and intermittency independently (**Figure 6B**). This suggested that this membrane potential could be used to navigate environments where only one of intermittency or frequency is high. To test this, we considered a navigator that used the difference between the membrane potential and its resting potential (i.e., $E_{leak}$) to generate an upwind bias:

$$p\left(\text{turn upwind}\middle|\text{turning}\right) = \frac{1}{1+e^{-g_{PN}(V-E_{leak})}} \tag{12}$$

where $g_{PN}$ is the base gain for this model chosen analogously to the other base gains (see Materials and methods). While the circuit-inspired model was outperformed by the single-sensor models when these were used in matching environments (i.e., the *F* model in the high-frequency plume and the *I* model in the high-intermittency plume) (**Figure 6C and D**), it performed better than the individual *F* and *I* models when those were used in suboptimal environments. Thus, the dual-sensing capability of the ORN-PN circuit translates directly to more effective navigation across diverse plumes. Of course, as our results above showed, some degree of modulation of the gains could further enhance performance (**Figures 4I and 6C and D**, purple) – say by amplifying frequency sensing in certain plumes. It would be interesting to investigate whether any such modulation is enacted by the insect olfactory circuit.

## Discussion

In this work, we used numerical simulations to explore the value of two temporal features of the signal – odor intermittency and encounter frequency – in navigating naturalistic odor plumes spanning a range of spatial and temporal complexity. These two features are a natural set in that they can be varied independently to create a variety of odor signals (**Figure 1**). Other complementary and complete quantities could be used, such as whiff and blank duration (**Rigolli et al., 2021**), but we focused on these since they are directly implicated by various experiments in walking *D. melanogaster*. The navigation model we proposed reduces two experimentally informed models of fly olfactory navigation into elementary transformations that separately extract odor intermittency and encounter frequency, and then uses these two 'sensors' to bias the agent upwind. Our model is phenomenological, exploring the utility of different odor signal features in different environments, and so does not necessarily implicate any particular neural architectures. An interesting finding here is that the optimal agent in the two simulated plumes assigned weights to the sensors that resembled the weights inferred from experiment (**Demir et al., 2020**; **Figure 4G and H**, Materials and methods). This suggests that the manner in which temporal features are extracted and processed within the *Drosophila* olfactory circuit may already be adapted to natural plume environments.

Our work explores normative strategies, so our results have no bearing on whether such adaptation actually occurs. There is, however, evidence that such adaptation may exist at the level of individual neurons: for example, moth ORNs adjust their encoding efficiency to the local statistics of pheromones (**Levakova et al., 2018**). Additionally, upwind orientation was found to be independent of intermittency for fixed frequencies (**Demir et al., 2020**), suggesting that such adaptation of sensor weight may actually be present in walking *Drosophila*. Our work suggests future experiments, based on simple modifications of existing experimental paradigms, that could be used to quantify this slower-scale adaptation. One could present the complex odor plumes we generated in our recent work (**Demir et al., 2020**), while modulating the overall statistics on a slower scale via the speed or strength of the upwind lateral perturbations, the wind speed, or both, and record how upwind orientation depends on frequency or intermittency. Additionally, in general, flying flies are more likely to experience more complex, high-frequency odor environments than walking flies due to flying flies being far from solid boundaries (**Connor et al., 2018**). Thus, if such modulation of sensor weight occurs, flying flies might naturally assign more weight to frequency sensing, which could be tested experimentally in wind tunnels for flight (**van Breugel and Dickinson, 2014**).

A key finding here is that the known circuitry of the *Drosophila* olfactory periphery, namely, in ORNs (*Gorur-Shandilya et al., 2017*; *Nagel et al., 2015*) and PNs (*Nagel et al., 2015*), responds to both odor intermittency and frequency, aiding robust navigation across many odor environments. This suggests that the known neural circuitry at the first stages of olfactory processing is tuned, to some degree, to naturalistic navigation tasks. In our simulations, this model is still suboptimal, and performance might be improved by including the effect of lateral inhibition, which has been shown to modulate the frequency range encoded by PNs (*Nagel et al., 2015*), as well as further processing in later stages of the circuit (*Rapp and Nawrot, 2020*). Also, we did not include much slower adaptive components (~10 s) of synaptic depression that modulate activity of *Drosophila* PNs (*Martelli and Fiala, 2019*). Given that this timescale is similar to that of the behavioral adaptation found by *Álvarez-Salvado et al., 2018*, it is plausible that this modulation could improve navigation. It has also been shown that knockdown of the priming factor unc13A impedes fast components of ORN-PN synaptic transmission in *Drosophila* (*Fulterer et al., 2018*; *Pooryasin et al., 2021*) and affects behavioral responses to signals at higher frequencies (*Fox and Nagel, 2021*). It would be illuminating to test how unc13A knockdown affects navigation in complex plumes of different frequency content.

In the latter half of this study, we simulated a variety of odor plumes using a simple drift-diffusion model (*Farrell et al., 2002*). A more precise approach would be to numerically integrate the Navier–Stokes equations describing the wind flow, together with advective-diffusive scalar transport describing the dispersion of a scalar concentration field (*Rigolli et al., 2021*). In such simulations, resolving odor concentrations to the viscous scale is very computationally expensive. This would likely preclude the investigation over more than a handful of distinct odor plumes, as our simplified model allowed us to explore here. On the other hand, such detailed simulations show that even in a single plume the statistics of the odor change significantly with distance from the source, and therefore animals may benefit from modulating sensory strategies during navigation (*Rigolli et al., 2021*). This is consistent with our finding that frequency sensing contributes more near the edges of the plume than it does near the centerline, and vice versa for intermittency sensing.

There are several aspects of olfactory navigation not considered in this work. In particular, we have neglected the role of bilateral sensing between the two antennae. In insects, bilaterally resolved concentration sensing has been demonstrated in flies (*Gaudry et al., 2013*) and implicated in navigation of laminar ribbons (*Duistermars et al., 2009*). Bilateral sensing has also been demonstrated in mice (*Rajan et al., 2006*), sharks (*Gardiner and Atema, 2010*), and even humans (*Wu et al., 2020*), and has been implicated in effective navigation in aquatic environments (*Michaelis et al., 2020*). Spatially resolved information has been shown theoretically to provide more information about an agent's position relative to the source of the odor (*Boie et al., 2018*) and aid olfactory navigation strategies, even in plumes with elements of stochasticity and turbulence (*Hengenius et al., 2021*). For very closely spaced antennae as in flies (<1 mm), these gradients are very difficult to resolve and so are often not useful for navigation (*Celani et al., 2014*; *Crimaldi and Koseff, 2001*; *Shraiman and Siggia, 2000*). Nonetheless, it would be interesting to consider the effect of bilateral comparisons of intermittency and frequency, particularly when modeling the navigation of species with larger antennae.

To this end, it has already been shown that bilateral comparisons of frequency allow agents to track the edges of some turbulent odor plumes (*Michaelis et al., 2020*). Additionally, recent work (*Rigolli et al., 2021*) has shown that odor intensity and temporal statistics are more useful in the central and outer regions of a turbulent plume, respectively, for predicting distance to the source. It is possible that in high-intermittency plumes organisms might use frequency to track the edges of odor plumes or even execute offset responses, such as those detailed in *Álvarez-Salvado et al., 2018*. Moreover, it has recently been shown that flies can use bilateral information to detect the direction of motion of odor signals (*Kadakia et al., 2021*), and that this information is particularly relevant in turbulent environments. In more diffuse and smooth plumes, odor velocity is less well-defined, and might be of more limited use. An interesting extension would be investigating how odor velocity could be incorporated optimally with odor intermittency and frequency in effective navigation.

For the sake of simplicity, we considered a model where agents move with a constant speed and only change orientation through a discretized turning paradigm, suggested by *Demir et al., 2020*. However, more diverse actions such as stopping and walking (*Demir et al., 2020*), speed modulation (*Álvarez-Salvado et al., 2018*; *Mafra-Neto and Cardé, 1994*), continuous heading modulation (*Álvarez-Salvado et al., 2018*), and casting/counter-turning behavior *Álvarez-Salvado et al., 2018*;

*Budick and Dickinson, 2006*; *Mafra-Neto and Cardé, 1994*; *Pang et al., 2018*; *Vickers and Baker, 1994* have also been observed in insect olfactory navigation. In future work, it will be worth investigating the role of intermittency and frequency in modulating behaviors such as these in different environments.

Finally, we have not explored the role of learning. The frequency and intermittency filters we used had no timescale longer than a few seconds, precluding history-dependent behavioral effects over longer timescales. History dependence in navigational decisions has been observed in flying fruit flies (*Pang et al., 2018*), where the magnitude of fly turns decreased with the number of signal encounters, in desert ants (*Buehlmann et al., 2015*), where ants used the existence of previously learned olfactory cues to navigate in a new environment, and in mice (*Gire et al., 2016*), where gradient climbing was abandoned for foraging when mice were sufficiently conditioned on known odor locations. Theoretical strategies such as infotaxis, where agents navigate by using cues to learn an internal probabilistic representation of their environment (*Vergassola et al., 2007*), also have some support in experiment (*Calhoun et al., 2014*; *Pang et al., 2018*). We find that robust navigation is enhanced by modulating intermittency and frequency sensing in time, and incorporating history dependence in our models could be done straightforwardly, with a few added parameters. Pairing this with behavioral experiments of the type suggested above would provide a fruitful direction for future study.

## Materials and methods
### Simulating ON and F responses to square waves

The frequency response function is defined as the convolution between the whiff onset time series $w(t)$ and an exponential filter with decay timescale $\tau_F$ where the whiff time series is a sum of delta functions occurring at the onset of each whiff. Thus, we have

$$F(t) = \int_{-\infty}^{t} w(t-s) e^{\frac{-s}{\tau_F}} ds = \sum_k \int_{-\infty}^{t} \delta(t-t_k-s) e^{\frac{-s}{\tau_F}} ds = \sum_k e^{\frac{-t-t_k}{\tau_F}} \tag{13}$$

where $k$ enumerates the whiffs. Note that $F(t+\Delta t) = F(t) e^{\frac{-\Delta t}{\tau_F}}$. Therefore, in discrete time steps we have $w(t+\Delta t) = 1$ if $odor(t) < K$ and $odor(t+\Delta t) \geq K$ and $0$ otherwise and $F(t+\Delta t) = F(t) \cdot e^{\frac{-\Delta t}{\tau_F}}$ if $w(t+\Delta t) = 0$ and $F(t+\Delta t) = F(t) \cdot e^{\frac{-\Delta t}{\tau_F}} + 1$ if $w(t+\Delta t) = 1$.

For $ON(t)$, we use Euler's method to numerically integrate *Equation 2* to obtain $A(t)$ and then similarly integrate the following equation:

$$\frac{dON}{dt} = \frac{1}{\tau_{ON}} (C(t) - ON(t)) \tag{14}$$

where $C(t)$ is defined in *Equation 1*, and the above equation is equivalent to *Equation 3*. $\tau_F$ was set to 2 s (*Demir et al., 2020*) while $\tau_A$ and $\tau_{ON}$ were set to 9.8 s and 0.72 s, respectively (*Álvarez-Salvado et al., 2018*). The detection threshold was assumed to be below the signal amplitude, and $k_d$ was set to be 1% of the signal amplitude.

### Calculation of ON, I, and F responses to square waves

To illustrate how the ON and F filters respond to the frequency and duration of odor signals, we consider their response to square-wave odor pulses of given frequency $f$, duration $D$, and amplitude $S_0$. We first consider the ON response. To understand the ON response, we first calculate $A(t)$. From *Equation (2)*, we have

$$\frac{dA}{dt} = \frac{1}{\tau_A} \cdot (odor - A) \tag{15}$$

Let $A_n$ denote the value of $A$ at the *offset* of the $n$th pulse of signal and $A_n^*$ denote the value of $A$ at the *onset* of the $n$th pulse. We wish to obtain a recursive relation for $A_n$, which will allow us to solve for $A_n$ and from there obtain the value of $A$ at all times. At the offset of a pulse, $odor = 0$ and $A$ will exponentially decay with time scale $\tau_A$ until the onset of the next pulse. This time of decay is given by $\frac{1}{f} - D$. Hence at the onset of the next pulse, $A_{n+1}^* = A_n \cdot e^{-\left(\frac{1}{\tau_A} \cdot \left(\frac{1}{f} - D\right)\right)}$. At this point, for a time period $D$, that is, until the offset of the $(n+1)$th pulse, $A$ obeys the equation

$$\frac{dA}{dt} = \frac{1}{\tau_A} \cdot (S_0 - A) \tag{16}$$

with initial value $A_{n+1}^*$. Hence,

$$\int_{A_{n+1}^*}^{A_{n+1}} \frac{dA}{S_0 - A} = \frac{D}{\tau_A} \tag{17}$$

and therefore, after substituting $A_{n+1}^* = A_n \cdot e^{-\left(\frac{1}{\tau_A} \cdot \left(\frac{1}{f} - D\right)\right)}$

$$A_{n+1} = A_n e^{\frac{-1}{f \tau_A}} + S_0 \left(1 - e^{\frac{-D}{\tau_A}}\right) \tag{18}$$

One can thus see that

$$A_n = A_0 e^{\frac{-n}{f \tau_A}} + S_0 \left(1 - e^{\frac{-D}{\tau_A}}\right) \sum_{k=0}^{n-1} e^{\frac{-k}{f \tau_A}} \tag{19}$$

$$= A_0 e^{\frac{-n}{f \tau_A}} + S_0 \left(1 - e^{\frac{-D}{\tau_A}}\right) \cdot \frac{1 - e^{\frac{-n}{f \tau_A}}}{1 - e^{\frac{-1}{f \tau_A}}}. \tag{20}$$

Once the number of pulses $n$ is much greater than $f\tau_A$, that is, $t \gg \tau_A$, we get

$$A_n \approx \frac{S_0 \left(1 - e^{\frac{-D}{\tau_A}}\right)}{1 - e^{\frac{-1}{f \tau_A}}}. \tag{21}$$

Since this is the value of $A(t)$ at the end of a pulse, it will be the maximum value of $A(t)$ over one period. Ultimately, however, we are interested in computing $ON(t)$, which obeys the equation

$$\frac{dON}{dt} = \frac{1}{\tau_{ON}} \cdot \left(\frac{odor}{odor + kd + A(t)} - ON\right). \tag{22}$$

To understand the response of $ON$, we can consider three different signal timescales. If the signal fluctuates quickly with respect to $\tau_A$, that is, $D$ and $\frac{1}{f} - D \ll \tau_A$, then for $t \gg \tau_A$ one can approximate $A(t)$ with its average value over one period, which is given by

$$f \cdot \left[\int_0^{\frac{1}{f} - D} \frac{S_0 \left(1 - e^{\frac{-D}{\tau_A}}\right)}{1 - e^{\frac{-1}{f \tau_A}}} \cdot e^{\frac{-t}{\tau_A}} dt + \int_0^D \frac{S_0 \left(1 - e^{\frac{-D}{\tau_A}}\right)}{1 - e^{\frac{-1}{f \tau_A}}} e^{\frac{-\left(\frac{1}{f} - D\right)}{\tau_A}} \cdot e^{\frac{-t}{\tau_A}} S_0 \cdot \left(1 - e^{\frac{-t}{\tau_A}}\right) dt\right] \tag{23}$$

$$= S_0 \cdot f \cdot D \tag{24}$$

Notice $f \cdot D = I$, the intermittency of the signal. Hence in this limit, and assuming $S_0 \gg kd$, when the signal is present, we have

$$\frac{dON}{dt} = \frac{1}{\tau_{ON}} \cdot \left(\frac{1}{1 + I} - ON\right) \tag{25}$$

Thus, $ON(t)$ obeys the same dynamics as $A(t)$, except that it adapts to a square wave of amplitude $\frac{1}{1+I}$ instead of $S_0$ and with a different timescale. Thus by the same reasoning as for $A(t)$, the maximum value of $ON(t)$ over one period (once $t \gg \tau_A, \tau_{ON}$) is approximately $\frac{1}{1+I} \cdot \frac{1 - e^{\frac{-D}{\tau_{ON}}}}{1 - e^{\frac{-1}{f \tau_{ON}}}}$, and the average value over one period is $I \cdot \frac{1}{1+I}$.

If instead $\tau_A \approx D$ or $\tau_A \ll D$, then $A(t) \approx odor(t)$, and we get

$$\frac{dON}{dt} = \frac{1}{\tau_{ON}} \cdot \left(\frac{1}{2} - ON\right) \tag{26}$$

and the average value of $ON(t)$ becomes $I/2$. (The maximum value would be $\frac{1}{2} \cdot \frac{1 - e^{\frac{-D}{\tau_{ON}}}}{1 - e^{\frac{-1}{f \tau_{ON}}}}$.)

Finally, we can consider the case where $\tau_A \gg D$ and $\tau_A \ll \left(\frac{1}{f} - D\right)$. In this case, $A(t) \approx 0$ and $ON(t)$ adapts to a square wave with amplitude $\approx 1$. The average value of $ON(t)$ is $I$ (and the maximum value would be $\frac{1-e^{\frac{-D}{\tau_{ON}}}}{1-e^{\frac{-1}{f\tau_{ON}}}}$).

In summary, we see that in all these cases the average value of $ON$ depends only on the intermittency and increases monotonically with intermittency.

For $F$, it is easiest to consider $F_n$ as the value of $F$ just after the *onset* of the $n$th pulse. Since $F$ increases by 1 at the onset of each pulse and then decays exponentially with timescale $\tau_F$ until the onset of the next pulse, one has

$$F_{n+1} = F_n \cdot e^{\frac{-1}{f\tau_F}} + 1. \tag{27}$$

Hence,

$$F_n = F_0 \cdot e^{\frac{-(n-1)}{f\tau_F}} + \frac{1-e^{\frac{-n}{f\tau_F}}}{1-e^{\frac{-1}{f\tau_F}}} \tag{28}$$

where $F_0$ is the value of $F$ right before the onset of the first pulse. For $t \gg \tau_F$, we have $n \gg f\tau_F$ and $F_n \approx \frac{1}{1-e^{\frac{-1}{f\tau_F}}}$. Since $F$ jumps at the onset of a pulse and then decays, this is the maximum value of $F$. The average value of $F$ over one period is thus

$$\frac{1}{1-e^{\frac{-1}{f\tau_w}}} \cdot f \cdot \int_0^{\frac{1}{f}} e^{\frac{-t}{\tau_w}} dt = f \cdot \tau_w \tag{29}$$

Hence, the average value of $F$ is linearly proportional to the frequency of the signal.

In a square wave, the $I(t)$ filter obeys the exact same dynamics as $A(t)$, except with a pre-factor of 1/2 (assuming the amplitude of the wave is above the detection threshold) and thus has an asymptotic average response of $I/2$.

## Connection of navigation model to logical gates

We claim that *Equation 5* is very similar to an OR gate in the variables $g_I ON$ and $g_F F$. To see this, let us first define what we mean by an OR gate. Normally, an OR gate in two binary variables A and B returns a 1 if any one of A, B is nonzero. This results in the following 'truth table':

## Standard OR gate

| A | B | Output |
|---|---|--------|
| 0 | 0 | 0 |
| 1 | 0 | 1 |
| 0 | 1 | 1 |
| 1 | 1 | 1 |

and can be expressed algebraically as $A + B - AB$. In our case, however, we want a null output to result in ½ since this should be the probability of turning upwind when no signal is present. Similarly, our variables of interest are $g_I ON$ and $g_F F$, which are nonbinary and in principle unbounded. Since in general we will want null outputs to be ½ and full outputs to be 1, it is natural instead to consider as variables A and B sigmoidal transformations of $g_I ON$ and $g_F F$. Thus, we can define for our purposes

$$A = \frac{1}{1+e^{-g_I ON}} \tag{30}$$

$$B = \frac{1}{1+e^{-g_F F}} \tag{31}$$

Then the truth table of an OR gate would look like the following table:

## Navigation model OR gate

| A | B | Output |
|---|---|---|
| 1/2 | 1/2 | 1/2 |
| 1 | 1/2 | 1 |
| 1/2 | 1 | 1 |
| 1 | 1 | 1 |

We then wish to determine an algebraic combination of A and B that will result in this output. Like in the case of a standard OR gate, it is easy to see we must go to second-order expressions in A and B. Due to the symmetry of the output in A and B, we need only consider symmetric second-order expressions:

$$output = a_0 + a_1 \left(A + B\right) + a_2 \left(A \cdot B\right) + a_3 \left(A^2 + B^2\right)$$

(32)

This gives us four equations with four unknowns (one equation for each row of our table), but one can see that the middle two equations are redundant and thus we have a free variable. One can thus set $a_3 = 0$ and get as an OR gate in our case:

$$output = -1 + 2 \left(A + B\right) - 2AB$$

(33)

In other words, for a full logical OR function we would have

$$p \left(\text{turn upwind} \big| \text{turning}\right) = -1 + 2 \left(\frac{1}{1+e^{-x}} + \frac{1}{1+e^{-y}}\right) - 2 \cdot \frac{1}{1+e^{-x}} \cdot \frac{1}{1+e^{-y}}$$

(34)

where we have defined $x = g_I ON$ and $y = g_F F$. With this definition, **Equation 5** then reads

$$p \left(\text{turn upwind} \big| \text{turning}\right) = \frac{1}{1+e^{-(x+y)}}$$

(35)

Comparing the two expressions, one can show numerically that they differ by at most 0.025, meaning for any $ON$ and $F$ values, $p \left(\text{turn upwind} \big| \text{turning}\right)$ for a true OR gate and for our model will differ by at most 2.5%. Hence, we claim that our model is a simple expression that well-approximates an OR gate. One can analogously compute what an AND gate would look like in our framework, giving

$$p \left(\text{turn upwind} \big| \text{turning}\right) = 1 + 2AB - \left(A + B\right)$$

(36)

We simulated agents in the video plumes using this strategy as well, and unsurprisingly, they performed poorly in both plumes. The performance in the high-frequency plume was slightly worse than the performance of the intermittency-only model in that plume, and the performance in the high-intermittency plume was slightly worse than that of the frequency-only model in that plume.

### Agent-based simulation in recorded odor plumes

The first plume recording we used is the same as used in *Álvarez-Salvado et al., 2018*. We call this plume the high-intermittency plume. The odor detection threshold of the agents was set by analyzing the signal in a region outside the plume. In this region, pixel values of 0 were removed and nonzero values were fit to a Gaussian. The detection threshold was then set to be the 3 standard deviations above the mean of this fit. 10,000 agents were initialized with uniformly distributed starting position, where the x-position was between 50 mm and 300 mm from the source and the y-position went from 80 mm below the source to 80 mm above the source. The initial heading angle was uniformly distributed from 0 to 360°. The simulation was run for the length of the video (240 s), and the discrete time step was set to be the reciprocal of the frame rate (1/15 s).

The second plume recording we used was taken from *Demir et al., 2020*. We call this the high-frequency plume. The odor detection threshold of each agent was set the same way it was in *Demir et al., 2020*. Again 10,000 agents were initialized with uniformly distributed initial position and

heading. The initial x-position was between 38.45 mm and 288.45 mm, and the initial y-position was between –74 mm and 86 mm. Initial heading was uniformly distributed from 0 to 360°. The simulation was run for 123.3 s, starting from the 600th frame of the video to the last frame, at 89.94 frames/s, corresponding to the frame rate used in *Demir et al., 2020*. The first 600 frames were dropped so that the plume had expanded to full size when the simulations began.

In both simulations, odor signal was computed by averaging over an elliptical antenna-sensing region in front of the agent, as in *Demir et al., 2020*. The length of the region's major axis was 1.5 mm, and the length of the minor axis was 0.5 mm. The ellipse was centered 1 mm in front of the agent. For all models, odor values below the detection threshold described above were set to 0 to minimize the effect of camera shot noise. When computing the *ON* filter, the $k_d$ value was also set at this detection threshold value. If agents went outside the frame region, then they were allowed to continue but received zero signal in those regions. Thus, there were no walls in these simulations.

For these simulations, $F$ was computed as for the square-wave pulses, with a detection threshold as described above, but we also enforced that the whiff time series $w(t)$ could not register two whiffs less than 40 ms apart to capture the idea that the time resolution of individual whiffs is not arbitrarily precise and to avoid spurious detections due to the random fluctuations in the signal, as suggested by *Demir et al., 2020*.

### Determination of base gains from experiment

The base gains, $g_{I0}$ and $g_{F0}$, which were used for the simulations in *Figure 3*, and in multiples of which the gains in *Figures 4 and 5* are reported, were determined the following way. *Demir et al., 2020* experimentally extracted a sigmoidal turning bias, as in *Equation 6*, except only using the $F$ filter and reported a gain of 0.242. We thus set $g_{F0} = 0.242$. $g_{I0}$ was set so that the contribution from $I$ in the high-intermittency plume would be roughly the same size as the contribution from $F$ in the high-frequency plume. So defining $I_0$ and $F_0$ to be typical $I$ and $F$ values in the high-intermittency and high-frequency plumes, respectively, we have $g_{I0}I_0 = g_{F0}F_0$. We thus determined a $g_{I0}$ of 1.936. For the PN model, we considered $V_0$ to be the average value of the membrane potential in a high-intermittency environment and then set $g_{PN}(V_0 - E_{leak}) = g_{F0}F_0$, where $E_{leak}$ was set to –70 mV (see below). We thus determined $g_{PN}$ to be 0.057 /mV. Finally, for the parameters dictating the navigational actions, the turn rate was set to 1.3 /s, walking speed to 10.1 mm/s, and filter decay timescale $\tau$ to 2 s, all in accordance with the findings of *Demir et al., 2020*. Note that the same timescale was used for the $I$ and $F$ filters.

### Statistical methods

Error bars for success rates (*Figure 3C*) were computed by bootstrapping data from a simulation of 10,000 flies – 1000 resamples were used with each resample size being equal to 10,000. Similarly, for the histograms of successful initial conditions, the data was resampled 1000 times, where each resample size was the size of the original data and means and standard deviations were computed and used for each histogram bin.

### Agent-based simulation in simulated odor plumes

The simulated odor plumes were created using the strategy laid out by *Farrell et al., 2002*. Plumes consisted of growing Gaussian packets of odor concentration, released as a Poisson process with rate $\lambda$, that were advected by a uniform mean wind velocity and perturbed by turbulent diffusivity. The concentration at a point $(x, y)$ due to a packet centered at $(x_i, y_i)$ was computed as

$$odor_i(x, y) = \frac{C_0}{\pi(R_0^2 + 4Dt_i)} exp\left(\frac{-r_i^2}{(R_0^2 + 4Dt_i)}\right), \tag{37}$$

where $r^2 = (x - x_i)^2 + (y - y_i)^2$, $R_0$ is the initial packet radius, $t_i$ is the time since the release of this particular packet, $D$ is a diffusivity that governs the packet growth, meant to account for molecular diffusivity and the effects of small eddies, and $C_0$ sets the initial concentration amplitude. The total $odor(x, y, t)$ is then the sum over all packets that have been released up to time $t$. The packet center was computed the following way:

$$x_i\left(t + \Delta t\right) = x_i\left(t\right) + U\Delta t + \eta_1 \tag{38}$$

$$y_i\left(t + \Delta t\right) = y_i\left(t\right) + \eta_2, \tag{39}$$

where $U$ denotes the mean wind velocity, and $\eta_1$ and $\eta_2$ are Gaussian white noise perturbations with mean 0 and standard deviation $\sqrt{2\kappa\Delta t}$ , representing the effects of turbulent dispersion with eddy diffusivity $\kappa$.

In general, parameters were chosen to be physically realistic and also give concentration time series and odor plumes that were qualitatively similar to those in the videos. To set $C_0$ , we defined the detection threshold to be 1 and enforced that an agent more than 1.6 standard deviations away from an initial packet would not be able to detect its presence. See the following table:

| Parameter | Explanation | Value |
|---|---|---|
| $U$ | Wind speed | $36-300\,\text{mm/s}$ |
| $D$ | Packet growth diffusivity | $10-52\,\text{mm}^2/\text{s}$ |
| $\kappa$ | Eddy diffusivity | $1000\,\text{mm}^2/\text{s}$ |
| $\lambda$ | Packet release rate | $5\,\text{Hz}$ |
| $R_0$ | Initial packet radius | $10\,\text{mm}$ |
| $C_0$ | Initial packet intensity | $3827.24\,(\text{a.u.})$ |
| $K$ | Odor detection threshold | $1\,(\text{a.u.})$ |

The order of magnitude for $D$ was set by the fact that attractive odorants for *D. melanogaster* tend to have molecular diffusivities of around $10\,\text{mm}^2/\text{s}$ , for example, ethyl acetate. The eddy diffusivity $\kappa$ was set in accordance with *Drivas et al., 1996*. The release rate and initial size were chosen to be similar to those in *Farrell et al., 2002*. The wind speed was chosen to be similar to those used experimentally in *Demir et al., 2020* and (*Álvarez-Salvado et al., 2018*).

Additionally, to improve computational efficiency, packets were no longer tracked once their $x$ position was so large that even if all released packets were at that position, the sum of their contributions would still be less than the detection threshold.

10,000 agents were initialized with uniformly distributed initial position and angle, with $x$ between 50 mm and 400 mm, $y$ between –110 mm and 110 mm, and $0° < \theta < 360°$, where $x$ and $y$ positions are defined relative to the source location, as in *Figure 3*. Plumes were simulated for enough time steps so that the expected $x$ position of a packet released at time 0 would be equal to the maximum initial $x$ for navigating agents, before navigating agents were introduced and simulated for 120 s. Once again, a trajectory's success was defined by whether it got within 15 mm of the source location.

To define the antenna-sensing region, space was discretized into 'pixels' with 0.154 mm as the pixel width, matching the spatial resolution of the high-frequency plume. The concentration was then computed by averaging over the pixels in an elliptical region, with the region defined as in the previous section.

To set the level of noise added to the $I$ and $F$ filters, we first computed a characteristic $I$ value in the simulated high-intermittency plume, $I_0$ , by averaging $I$ values over a region $192\,\text{mm} < x < 205\,\text{mm}$ and $0\,\text{mm} < y < 9\,\text{mm}$ and then averaging over the length of the simulation. We did the same for $F$ values in the simulated high-frequency plume to obtain $F_0$ . The values we obtained were $I_0 = 0.388$ and $F_0 = 3.14$. We then used 5% of these values as the standard deviation for Gaussian white noise to be added to the output of the $I$ and $F$ filters, respectively, at each time step. We also used $I_0$ and $F_0$ as representative $I$ and $F$ values in order to assign a single relative filter weight with which to color each set of gains in *Figure 4G*.

## Investigating the role of filter timescales

To understand how performance depended on the filter timescales $\tau_I$ and $\tau_F$ , we varied the two timescales independently, and for each pair of timescales simulated 10,000 flies in the two simulated plumes explored thus far. No noise was added to the sensor outputs, and gains were set at the base gains. Given that the average response of the intermittency filter is independent of the filtering timescale, it is unsurprising that for the fixed $\tau_F$ performance does not change significantly for values of $\tau_I$

nearly two orders of magnitude apart and only starts to degrade once the timescale gets on the order of 10 s (*Figure 4—figure supplement 2A and B*). This degradation is also expected: at very long timescales, it requires significant time for the $I$ filter to reach an appreciable value, even in the case of constant odor. There was also no significant difference in performance in either plume between an $I$-only model with an infinitely fast ($\tau_I = 0$) timescale (and thus flat response power spectrum) and an $I$-only model with a 2 s timescale. This is to be expected as even with an infinitely fast timescale such a model has an upwind bias if and only if the signal is present and thus is only responding to the intermittency of the signal. We also see that performance is impacted by varying $\tau_F$ (*Figure 4—figure supplement 2A and B*) but that this is largely equivalent to fixing $\tau_F$ but varying $g_F$ instead (*Figure 4—figure supplement 2C*), as predicted by *Equation 29*.

### ORN and PN circuit model

ORN firing rates were computed from *Equations 7 and 8*. Once odor activity $a$ was obtained, it was convolved with a normalized sum of two gamma distributions, $N \cdot (\Gamma_1 - 0.5 \cdot \Gamma_2)$, where the timescales for the two gamma distributions were 6 ms and 8 ms, respectively (*Gorur-Shandilya et al., 2017*), and the shape parameters 2 and 3, respectively, giving the shape seen in *Figure 6A*. This convolution was then multiplied by 300 Hz to get a firing rate. Since the model is only valid in regions where $K_{on} < odor < K_{off}$, we set any odor less than $K_{on}$ to 0. In the simulated plumes, $K_{on}$ was set to 1 and $K_{off}$ was set to 400. $a_0$ was set to 0.15 in order to get a baseline firing rate of about 40 Hz in the presence of continuous odor. In order to ensure the activity would go to 0 once there was no signal, $\epsilon_L$ was bounded below by $\epsilon_L$ and $\epsilon_L$ was set to be greater than the steady-state $\epsilon$ when no signal is present, which is given by $\ln\left(\frac{1}{a_0} - 1\right) \approx 1.73$. Thus, $\epsilon_L$ was set to 2.5 and activity less than $\frac{1}{1+e^{\epsilon_L}}$ was set to 0. $\beta$ was set to 0.8 /s, in accordance with *Gorur-Shandilya et al., 2017*.

Once the ORN firing activity was obtained, PN membrane voltages were obtained using *Equations 9–11*. All parameters in *Equations 9–11* were taken from *Nagel et al., 2015*. Since the fastest timescales were around 5 ms, responses were calculated through Euler integration with a timescale of 0.5 ms.

## Acknowledgements

We thank Mahmut Demir for providing measured plume data and advising on the simulations, Hope Anderson for help with agent-based simulation codes, and Henry Mattingly and Aarti Sehdev for helpful discussions. VJ and TE were supported by the Program in Physics, Engineering and Biology at Yale University. NK was supported by a postdoctoral fellowship through the Swartz Foundation for Theoretical Neuroscience, by postdoctoral fellowship NIH F32MH118700, and by postdoctoral fellowship NIH K99DC019397.

## Additional information

### Competing interests

Thierry Emonet: Reviewing editor, *eLife*. The other authors declare that no competing interests exist.

### Funding

| Funder | Grant reference number | Author |
| --- | --- | --- |
| National Institutes of Health | F32MH118700 | Nirag Kadakia |
| National Institutes of Health | K99DC019397 | Nirag Kadakia |
| Yale University | Program in Physics | Viraaj Jayaram |
| Sloan-Swartz Foundation | | Nirag Kadakia |
| Yale University | Program in Biology | Viraaj Jayaram |

| Funder | Grant reference number | Author |
|--------|------------------------|--------|
| Yale University | Program in Physics Engineering and Biology | Viraaj Jayaram |

The funders had no role in study design, data collection and interpretation, or the decision to submit the work for publication.

## Author contributions

Viraaj Jayaram, Conceptualization, Data curation, Formal analysis, Investigation, Methodology, Software, Validation, Visualization, Writing – original draft, Writing – review and editing; Nirag Kadakia, Conceptualization, Formal analysis, Funding acquisition, Investigation, Methodology, Software, Supervision, Validation, Visualization, Writing – original draft, Writing – review and editing; Thierry Emonet, Conceptualization, Formal analysis, Funding acquisition, Investigation, Methodology, Project administration, Resources, Supervision, Validation, Writing – original draft, Writing – review and editing

## Author ORCIDs

Viraaj Jayaram (iD) http://orcid.org/0000-0002-9607-2214
Nirag Kadakia (iD) http://orcid.org/0000-0001-9978-6450
Thierry Emonet (iD) http://orcid.org/0000-0002-6746-6564

## Decision letter and Author response

Decision letter https://doi.org/10.7554/eLife.72415.sa1
Author response https://doi.org/10.7554/eLife.72415.sa2

# Additional files

## Supplementary files

• Transparent reporting form

## Data availability

All data analyzed in this study are available from the original publications. Codes are available at https://github.com/emonetlab/plume-temporal-navigation (copy archived at swh:1:rev:eba94d35d4e5378d00ab7f03e528a24a67474d10).

The following previously published datasets were used:

| Author(s) | Year | Dataset title | Dataset URL | Database and Identifier |
|-----------|------|---------------|-------------|-------------------------|
| Demir M, Kadakia N, Anderson H, Clark D, Emonet T | 2021 | Data presented in "Walking *Drosophila* navigate complex plumes using stochastic decisions biased by the timing of odor encounters" | https://doi.org/10.5061/dryad.4j0zpc87z | Dryad Digital Repository, 10.5061/dryad.4j0zpc87z |
| Álvarez-Salvado E, Licata A, Connor E, McHugh M, King B, Stavropoulos N, Victor J, Crimaldi J, Nagel K, King B | 2019 | Data from: Elementary sensory-motor transformations underlying olfactory navigation in walking fruit flies | https://doi.org/10.5061/dryad.g27mq71 | Dryad Digital Repository, 10.5061/dryad.g27mq71 |

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
