## [Editor Report]

This article by Jayaram and colleagues uses computational modeling approaches to examine how temporal filtering of an odor signal contributes to navigation success in different odor environments. The article advances the literature in considering how different algorithms may be optimal for different environments. The provided evidence suggests an intriguing trade-off between frequency and ‘intermittency’ sensing.

---

## [Decision Letter]

**Decision letter after peer review:**

Thank you for submitting your article "Navigating a diversity of turbulent plumes is enhanced by sensing complementary temporal features of odor signals" for consideration by *eLife*. Your article has been reviewed by 3 peer reviewers, and the evaluation has been overseen by a Reviewing Editor and Ronald Calabrese as the Senior Editor. The following individual involved in review of your submission has agreed to reveal their identity: Jeff Riffell (Reviewer #1).

Essential revisions:

1. Discuss the role of odor intensity, for example as suggested by referee 1.

2. Test the performance of the algorithm to more realistic conditions either in vitro or in vivo (see suggestions referee 1).

3. Include a Figure like 2C and 2F for the simulated odor plumes considered in the second half of the paper.

4. Better justify the use of expression (6) for example as suggested by referee 2.

5. Show results for a logical 'or' combination of inputs and consider other alternatives, as suggested by referee 2.

6. Compare results to a null model that surges in response to any signal above threshold with no frequency filtering, and analyse performance as a function of the filtering cutoffs.

7. Discuss what you expect the frequency content of natural plumes to be.

*Reviewer #1 (Recommendations for the authors):*

I am enthusiastic about the authors past work in this domain, but I have a few suggestions on the current work.

1. Incorporating odor intensity into the simulated model. Presumably, the authors have related information on this topic from their previous work. I'm curious whether the interaction between "rate" and "intermittency" is related to the flux of the odor experienced by the navigating fly (or agent).

2. The authors simulate only two different plumes, but in nature, these plumes are dynamic and could be a combination based on the turbulent air conditions. Under a range of intermittency/rate conditions, how do the model results hold up? And how about then the plume dynamically changes between the two environments?

3. Alternately, empirical testing using fruit flies in an arena like their previous study, but stimulating the flies with odor pulses that simulated the model intermittent and rate plumes, would strengthen the authors' results.

*Reviewer #2 (Recommendations for the authors):*

I have three suggestions for the authors:

1. It would be very helpful to have a Figure like 2C and 2F for the simulated odor plumes considered in the second half of the paper in oredr to make a comparison.

2. The expression (6) while sensible does not have a strong justification. The authors should clarify if there is any indication that the neural system of flies is actually combining the output of two different signal processing units, one for intermittency and one for frequency, through a simple perceptron as they do in (6). It would also be interesting to know if there are ways to test this hypothesis.

3. Related to 2, a different and possibly more compelling way to express the combination of inputs is to use a simple perceptron that implements an OR function instead of (6). Indeed, this appears the most natural choice to obtain high upwind turning probabilities with high intermittency or high frequency. The authors are warmly encouraged to show results for this other choice, and possibly other "logical" ANNs as well.

*Reviewer #3 (Recommendations for the authors):*

line 283: setting sensor gains to zero. Can this be compensated by increasing the gain of the other sensor?

~line 334: what is the molecular diffusivity of air how does this compare to the values modeled here?

Figure 4, E and F: it might be helpful to explore a finer range of gains close to 1 here.

---

## [Author Response]

Essential revisions:1. Discuss the role of odor intensity, for example as suggested by referee 1.2. Test the performance of the algorithm to more realistic conditions either in vitro or in vivo (see suggestions referee 1).3. Include a Figure like 2C and 2F for the simulated odor plumes considered in the second half of the paper.4. Better justify the use of expression (6) for example as suggested by referee 2.5. Show results for a logical 'or' combination of inputs and consider other alternatives, as suggested by referee 2.6. Compare results to a null model that surges in response to any signal above threshold with no frequency filtering, and analyse performance as a function of the filtering cutoffs.7. Discuss what you expect the frequency content of natural plumes to be.

Thank you for the close reading of our manuscript and the valuable feedback. We have addressed all of these points through revisions, as outlined in point-by-point detail below.

Reviewer #1 (Recommendations for the authors):I am enthusiastic about the authors past work in this domain, but I have a few suggestions on the current work.1. Incorporating odor intensity into the simulated model. Presumably, the authors have related information on this topic from their previous work. I'm curious whether the interaction between "rate" and "intermittency" is related to the flux of the odor experienced by the navigating fly (or agent).

We agree that a proper consideration of the role of intensity would strengthen the work. Instead of simply binarizing the odor signal, we have now treated the odor signal directly, and filtered it using an adaptive compression module, as done in previous work (Alvarez-Salvado et al., *eLife* 2018; Gorur-Shandilya, Demir et al., *eLife* 2017). See changes to Figure 3. We also included the roles of signal intensity and adaptive compression in the new Figure 6, where we model a navigator that uses a biophysical implementation of the neuronal circuit in the fly olfactory periphery to detect odor signal. Regarding odor flux, Zhou and Wilson (*J. Neurosci.* 2012) have shown that *Drosophila* ORN responses are invariant to air speed. Thus, ORNs respond to changes in odor concentration rather than odor flux. We added a line in the introduction mentioning this result.

2. The authors simulate only two different plumes, but in nature, these plumes are dynamic and could be a combination based on the turbulent air conditions. Under a range of intermittency/rate conditions, how do the model results hold up? And how about then the plume dynamically changes between the two environments?

While we did use two measured plumes (Figures 2-3), we simulated a wide range of plumes (Figure 5), spanning a wide range of frequencies (0-5Hz) and intermittencies (0-0.9). Those results provided a wider degree of detail not studied in the initial explorations in Figures3 and 4, and allowed us to make broader conclusions about the tradeoff of intermittency and frequency sensing. Each of these plumes spans a broad range of temporal statistics, at different parts of the plume. We added Figures 4C-D to reflect this. Thus, agents experience a range of statistics while navigating a single environment.

3. Alternately, empirical testing using fruit flies in an arena like their previous study, but stimulating the flies with odor pulses that simulated the model intermittent and rate plumes, would strengthen the authors' results.

We wish to keep this a purely computational study, but we will keep this advice in mind to perform a separate, experimental study to test the findings presented here.

Reviewer #2 (Recommendations for the authors):I have three suggestions for the authors:1. It would be very helpful to have a Figure like 2C and 2F for the simulated odor plumes considered in the second half of the paper in oredr to make a comparison.

We have now added this data as Figure 4C and 4D.

2. The expression (6) while sensible does not have a strong justification. The authors should clarify if there is any indication that the neural system of flies is actually combining the output of two different signal processing units, one for intermittency and one for frequency, through a simple perceptron as they do in (6). It would also be interesting to know if there are ways to test this hypothesis.

To address this and similar comments from reviewer 3, we have now added a biophysical model of the fly olfactory sensory periphery, from odor transduction to ORN firing to PN response. Our model combines ORN models and ORN-PN synapse models that have been developed and justified in prior works (Gorur-Shandilya, Demir, et al., *eLife* 2017; Nagel and Wilson, *Nat. Neurosci.* 2015; Fox and Nagel, “Synaptic control of temporal processing in the *Drosophila* olfactory system” *arXiv* 2021). We find that this circuit naturally responds to both intermittency and frequency in a manner that is qualitatively similar to our simpler heuristic model. Further, agents navigating with this sensory module can navigate both high frequency and high intermittency plumes reasonably well – notably, better than only intermittency or only frequency sensing agents – in analogy to the performance of our combined model. We added new paragraphs in the results and Discussion sections about this new result.

3. Related to 2, a different and possibly more compelling way to express the combination of inputs is to use a simple perceptron that implements an OR function instead of (6). Indeed, this appears the most natural choice to obtain high upwind turning probabilities with high intermittency or high frequency. The authors are warmly encouraged to show results for this other choice, and possibly other "logical" ANNs as well.

Following the reviewer’s suggestion, we have now investigated the consequences of using different logical combinations of intermittency and frequency signals in Equation 5 (previously Equation 6). We have added a sub-section to our Materials and methods contextualizing Equation 5 in terms of an OR gate and discussing other logical transformations as well.

Reviewer #3 (Recommendations for the authors):line 283: setting sensor gains to zero. Can this be compensated by increasing the gain of the other sensor?

We addressed this in fuller detail in Figure 4, where we swept the gains of both sensors independently. In general, increasing the gain of one sensor cannot compensate for loss of performance by another, especially when noise is taken into account.

~line 334: what is the molecular diffusivity of air how does this compare to the values modeled here?

A typical odorant diffusivity is of order 10 mm^2^/s in air. See also lines 965-976 in Materials and methods, where we discuss and justify plume simulation parameter choices in more detail.

Figure 4, E and F: it might be helpful to explore a finer range of gains close to 1 here.

We did a finer sweep of the gains to generate the points in Figure 4I.